

# Assessing Soil and Potential Air Temperature Coupling Using PALM-4U: Implications for Idealized Scenarios

Patricia Glocke[1], Christopher C. Holst[2], Basit A. Khan[1,3], and Susanne A. Benz[1]

[1]Institute of Photogrammetry and Remote Sensing, Karlsruhe Institute of Technology, 76131 Karlsruhe, Germany
[2]Institute of Meteorology and Climate Research, Atmospheric Environmental Research, Karlsruhe Institute of Technology, 82467 Garmisch-Partenkirchen, Germany
[3]Mubadala Arabian Center for Climate and Environmental Sciences (ACCESS), New York University Abu Dhabi, Abu Dhabi, United Arab Emirates

**Correspondence:** Patricia Glocke (Patricia.Glocke@kit.edu)

**Abstract.** Underground heat extremes amplified by e.g., underground infrastructure or badly adjusted geothermal systems have long been discussed in geosciences. However, there is little emphasis on the exchange between these subsurface heat extremes and the atmosphere. To address the issue, this study investigates the impact of varying soil temperatures on potential air temperatures in an idealized domain using the turbulence and building resolving large eddy simulation urban micro-climate model PALM-4U. This involves two steps: first we test if and how idealized domains can be simulated, second the coupling between surface and subsurface energy fluxes or rather temperatures in air and soil are in focus. We develop several scenarios, distinguishing between cyclic or Dirichlet/radiation boundary conditions along the x-axis, between summer and winter, as well as between various land cover types. Our results demonstrate that cyclic boundary conditions induce modifications of the potential air temperatures due to changes in the soil temperature. The magnitude of the impact varies with respect to the tested land covers, which primarily affect absolute temperatures. Daytime and season have a larger influence on the magnitude of the modifications. A 5 K increase in subsurface temperatures at 2 m depth results in a maximum of a 0.38 K increase for near surface potential air temperatures in winter between 09:00 and 10:00 local time after three days of simulation. When soil temperatures are decreased, we find predominantly inverse patterns. The least influence is found during summer at 09:00 local time where the elevated soil temperatures increase potential air temperatures by only 0.02 K over short- and tall grass, and 0.18 K over bare soil. When using Dirichlet/radiation boundary conditions, the atmosphere cannot develop freely and changing soil temperatures do not impact potential air temperatures.

These results help to enhance our understanding of the coupling between soil- and atmospheric temperatures and also provide recommendations for the simulability of idealized but reality-oriented scenarios in PALM-4U. It is one of the first studies that demonstrates that heat and cold sources in the soil can affect atmospheric parameters.

## 1 Introduction

Anthropogenic encroachments in the environment lead to elevated temperatures (Intergovernmental Panel On Climate Change, 2023). More than 50% of the global population lives in cities and is exposed to a climate that is characterized by the built



environment (Oke et al., 2017; Chakraborty and Lee, 2019; Benz et al., 2021). This percentage is projected to increase (UN2, 2019). Urban heat islands affect the health, the general well-being and the productivity of the city dwellers (Tong et al., 2021; Shahmohamadi et al., 2011; Heaviside et al., 2017). Furthermore, with increasing temperatures in course of climate change and ongoing urbanisation, this issue exacerbates (Oke et al., 2017; Rizwan et al., 2008; Manoli et al., 2019; Huang et al., 2019). Accordingly, understanding the mechanics that drive temperatures in an altered environment is of uttermost importance, including their connection to the underground.

Anthropogenic activities alter subsurface temperatures from their natural state due to numerous heat sources and modified heat fluxes for example through urbanization processes (Tissen et al., 2019; Menberg et al., 2013). These induce rising temperatures in the soil and in the aquifer (observed in groundwater temperatures). Here, we use the term "soil temperatures" to describe temperatures at a shallow depth of 2 m and predominantly in the unsaturated zone. Numerous studies primarily within the groundwater research community have investigated the temperature in the subsoil and depicted multiple heat sources, especially in urban areas and the built environment, which impacts soil and groundwater temperatures (Epting et al., 2017; Noethen et al., 2023; Tissen et al., 2019). This can be measured worldwide in boreholes and groundwater wells. Anthropogenic heat is accumulated and stored in the subsurface beneath cities, contributing to altered heat fluxes and higher temperatures compared to the rural surroundings (Tissen et al., 2019; Benz et al., 2018). Especially heated buildings and underground infrastructure like tunnels, underground parking lots, subways, water pipes, sewers or basements lead to modifications in the thermal regime of the subsurface. Mostly, these constructions add energy to the system, resulting in elevated subsurface temperatures (Oke et al., 2017; Benz et al., 2022). Just a single construction in the subsurface can lead to a thermal pollution in the surrounding of several degree Celsius (Attard et al., 2016). For example, ground temperatures near underground parking garages can be up to 10 K warmer. Also, the ground temperature next to waste water pipes, mining or landfills differ between 3 K to 10 K compared to their surrounding. Hence, a highly developed buried infrastructure is chiefly responsible to subsurface warming (Benz et al., 2022; Böttcher and Zosseder, 2022). Not only anthropogenic but also natural sources like hot springs can significantly alter the ground temperatures (Tissen et al., 2019).

However, the thermal coupling between the underground and the atmosphere is complex. Meteorological forcing, soil moisture, soil temperature, the heat transfer within the subsurface, and vegetation influence the land-atmosphere flux exchanges (Gao et al., 2008). The release of heat from the subsurface to the atmosphere can modify boundary layer dynamics, local wind patterns, and affect atmospheric stability (Rahman et al., 2015; Wouters et al., 2019; Brunsell et al., 2011). These atmospheric changes in turn can impact local meteorology and air quality (Hermoso de Mendoza et al., 2020; Asaeda and Ca, 1993; Santamouris et al., 2017).

Most land-atmosphere studies still focus on the impact of the atmosphere on the ground as a top-down scheme with a very simple treatment of the soil. For example, Staniec and Nowak (2016) employ a mathematical model to project annual soil temperature distributions, based on the transient heat conduction equation. They find significant sensitivity of soil temperatures to alterations in air temperature. Several studies model soil temperature derived from air temperatures with physical models or





with machine learning (Bayatvarkeshi et al., 2021; Hu et al., 2017; Liang et al., 2014). In addition, Taylor and Stefan (2009) analyze the potential impacts of climate and land use on surface and shallow groundwater temperatures with the aid of a 1-D

heat diffusion equation. Similar, Nitoiu and Beltrami (2005) depict subsurface thermal disturbances due to land use changes. Kurylyk and MacQuarrie (2014) examine the response of subsurface temperature to changes in atmospheric conditions with an analytical solution for a one-dimensional, transient conduction–advection equation and validated it with numerical methods.

In our study we ask the reverse: do alterations in soil temperatures impact potential air temperatures? To study the impact of subsurface heat on the near surface atmosphere with various seasons and land cover settings, we utilize the turbulence

and building resolving large eddy simulation urban climate model PALM-4U (Parallelized Large-Eddy Simulation Model for Urban applications) version 23.10 (Maronga et al., 2015). The PALM-4U model system uses large eddy simulations (LES) to calculate turbulent flows and incorporate both, subsurface and atmospheric processes. It can provide valuable insights into their complex dependencies. In order to gain a general understanding of how the underground temperature extremes impact the atmosphere, we model an idealized domain without any infrastructure or topography.

Accordingly, we set out to conduct a sensitivity analysis and address three distinct questions: (a) how to depict a realistic but idealized domain in PALM-4U?; (b) Do heat or cold extremes in the soil modify potential air temperatures?; and (c) what parameters affect these modifications?

To answer these questions we run our model for a total of 36 scenarios based on different lateral boundary conditions, seasons, land covers, and subsurface temperatures (Fig. 1). To our knowledge we are the first to investigate this bottom-up procedure

which involves modifying soil temperatures and evaluating their effects on surface parameters, particularly atmospheric temperatures. Currently, there has been no quantitative assessment conducted. We aim to to enhance our incomplete understanding of the interplay between evolving thermal conditions in the subsoil and the atmosphere. Additionally, the research seeks to investigate the feasibility of simulations in an idealized scenario and elucidate its associated constraints. These idealized scenarios provide paramount understanding of the potential influence of varying soil temperatures in an unaltered environment.





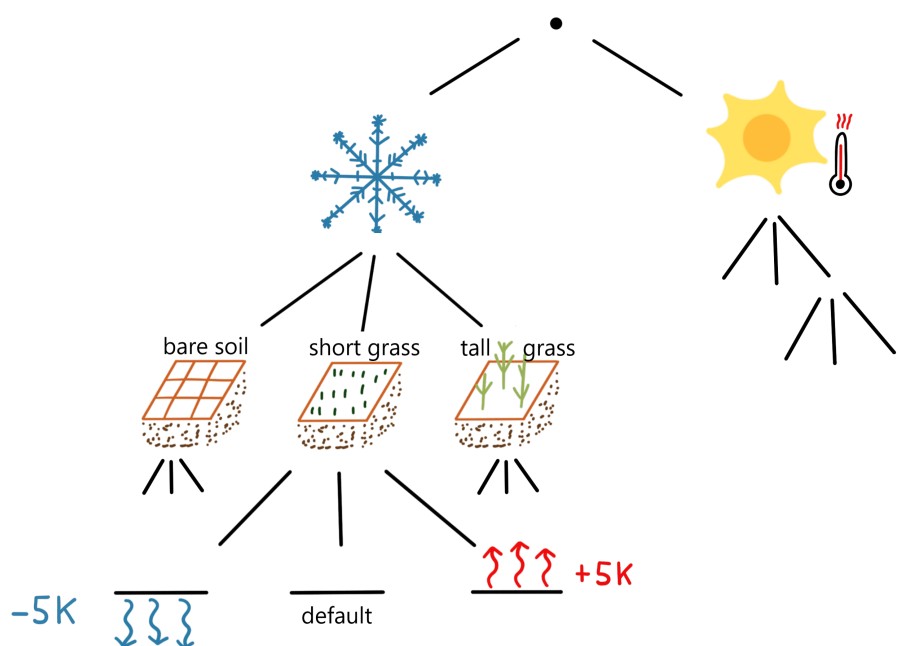

**Figure 1.** Overview over all simulated scenarios. The dot at the top of the figure stands for the used boundary condition i.e., cyclic or Dirichlet/radiation.

## 2  Materials and Methods

### 2.1  PALM-4U

In this study we use turbulence and building resolving LES urban climate model PALM-4U version 23.10. PALM-4U is able to simulate meteorological processes at building-resolving scales, down to sub-meter scales. The model has been developed at the Institute of Meteorology and Climatology at the Leibniz Universität Hannover. The specifications and capabilities of the model are described in detail by Maronga et al. (2020). Here we employ the model to investigate the coupling of temperatures between the soil and the atmosphere, with a focus on deep soil - atmosphere interactions. The model considers only conductive heat transport i.e., no groundwater flow in the subsurface. As its deepest layer is at 2 m depth, we can assume it does not reach the aquifer in most places. Effective thermal conductivity within the simulated 2 m soil layer is estimated for each grid cell individually based on soil type, as specified by geostationary input, and soil moisture.

In order to produce realistic results, realistic initial and lateral boundary conditions for the atmosphere and the soil are required. The mesoscale numerical Weather and Research Forecasting (WRF) model (WRF Community, 2000; Skamarock et al., 2019) can be used to provide large-scale meteorological forcing like wind and temperature as profiles or 3-D data (Maronga et al., 2020). Furthermore, precise information on the geospatial components, the surface and local environment is





essential. Thus, PALM-4U incorporates also static surface information. The so-called static driver contains for example the topography, properties of the surface, geometries (height and type) of buildings and street canyons, soil type (e.g., coarse, medium, fine etc.), vegetation type, land cover (as short or tall grass, crops, shrubs, forests, bare soil, etc.), or the current state and type of the plant canopy. Variables like roughness lengths, emissivity, and leaf area index are automatically prompted. In addition, it is worth noting, that up to twenty different soil layers with corresponding temperature, moisture, and root fraction can be implemented.

## 2.2 Idealized domain

We aim to disentangle the effects of changes in soil temperatures on the atmosphere from the influence of surface hetero-geneities. For this purpose, an idealized test domain is set up: our domain contains no plants such as bushes or trees nor infrastructure like roads or houses. It is a homogeneous flat terrain. We apply the land surface model (LSM) (Gehrke et al., 2020) and the radiative transfer model (Krč et al., 2020) integrated in PALM-4U. The LSM serves for modeling surface fluxes i.e., the exchange of heat and moisture between the soil and the atmosphere, depending on the land cover (e.g., soil, water, pavement), the radiation budget of the surface, and the atmospheric conditions. The radiative transfer model calculates the radiation budget of the Earth's surface and deriving 3-D radiative interactions between surfaces and vegetation. The integrated models are based on the theory of a stable system i.e., no climate change and no artificial or natural (e.g., geothermal) heat sources (Gehrke et al., 2020; Krč et al., 2020).

## 2.3 Scenarios

For each season (summer and winter) and land cover type (bare soil, short grass and tall grass) we simulate multiple scenarios, adjusting the temperatures of the deepest soil layer (-2 m):

(i) Status Quo: with default temperatures derived from a two-year soil spin-up.

(ii) Heat source: the temperature of the deepest layer is increased by +5 K.

(iii) Cold source: In this scenario, we are examining the potential effects of cooling the soil on the parameters of study, we decrease the soil temperature of the deepest soil layer by -5 K.

Figure 1 shows the scheme of all scenarios, resulting in 18 runs for each of the two applied lateral boundary conditions. We set up the model to simulate three days with hourly output of the data. For the sake of not only examining the short-term behavior, but also the long-term developments, we additionally run selected scenarios with Dirichlet/radiation and cyclic boundary conditions for one year.

### 2.4 Initial and boundary conditions

Significant emphasis is placed on setting up the initial and lateral boundary conditions (LBC hereafter). The Fortran-namelist-input parameter (p3d) file is provided in the section "Code availability". The p3d file contains all the control parameters required



for the model, such as initial and boundary conditions, resolution, which variables are output after the simulation, etc. In detail, an idealized, blank domain is located at latitude = 52.5103, longitude = 13.1418, which corresponds geographically to an undeveloped area near Berlin. We chose 10 m isotropic grid spacing with 50 x 50 grid points in x and y direction (which corresponds to a 500 m x 500 m domain) as well as 50 grid points in the height z. We apply vertical stretching with a factor of 1.05 at a level of 300.0 and resulting in a maximum of $dz\_max = 50.0$. Thus, in total z is 670.052 m. We also tested our

simulations in a greater domain (2000 m x 2000 m x 4000 m) but found no impact of domain size on our results. To save computational costs we focus here on the simulations in a smaller domain.

We use cyclic boundary conditions along the y-axis. In our different simulations we tested different LBC along the x-axis which are (a) Dirichlet/radiation and (b) cyclic. In case (a) the Dirichlet condition is used at the inflow side for each variable. This means that a constant vertical profile is determined at the beginning and the inflow properties do not change with time.

At the outflow side, the radiation condition is employed across all velocity variables. The boundaries are not connected to each other. A turbulent flow develops by the inherent friction of the flow itself, as for example through irradiation and fluxes (Zhiyin, 2015). In case (b) under cyclic boundary conditions turbulence can be generated by itself along the x-axis. The turbulence does not experience any horizontal boundaries and hence it can develop freely, flow out and enter the domain again with its outflow values (PALM-4U Documentation).

Half of the simulations starts in summer (22.07.2022), whereas the remaining half starts in winter (13.01.2022), all at midnight. Those dates are chosen because they are the closest median winter and summer temperatures for Berlin in 2022 (calculated with data from German Weather Service, Climate Data Center DWD Deutscher Wetterdienst. Berlin has a temperate oceanic climate with moderate summers and mild winters. The Köppen climate type corresponds to Cfb (Kottek et al., 2006). The clear-sky scheme is used for the radiative forcing in the simulation, which disregards clouds, and modifications in aerosols.

The temperature gradients of the initial temperature profile (in K per 100 m) and the height levels above which the temperature gradient is effective can be seen in Table 1. This is derived from the $init\_atmosphere\_pt$ variable of a previous WRF simulation for Berlin. For the actual run of the idealized case no dynamic driver is used but only static variables (i.e., the static driver) and a p3d file.

**Table 1.** Vertical temperature gradient (K/100 m) for given heights.

| Heights (m) | Vertical Gradient (K/100 m) |
|:---:|:---:|
| 45 – 95 | 5.695 |
| 95 – 185 | 1.251 |
| 185 – 2448 | 0.339 |
| 2448 – 3888 | 0.258 |

Analogously, the WRF data also serves for the calculation of the humidity gradient, $q\_vertical\_gradient$. Here, the data of

the variable $init\_atmosphere\_qv$ in the middle of the domain at x=5000 and y=5000 are used. The levels ($q\_vertical\_gradient\_level$) are selected according the vertical profile. The calculated gradients (kg/kg per 100 m) between the heights (z) are the following:



**Table 2.** Vertical humidity gradient (kg/kg per 100 m) for given heights.

| Heights (m) | Vertical Gradient (kg/kg per 100 m) |
|---|---|
| 45 − 155 | $5.00 \cdot 10^{-4}$ |
| 155 − 275 | $1.70 \cdot 10^{-4}$ |
| 275 − 405 | $-0.71 \cdot 10^{-4}$ |
| 405 − 565 | $-3.10 \cdot 10^{-4}$ |
| 565 − 1599 | $-5.17 \cdot 10^{-4}$ |
| 1599 − 1630 | $2.31 \cdot 10^{-4}$ |
| 1630 − 2448 | $1.49 \cdot 10^{-4}$ |
| 2448 − 3888 | $-0.65 \cdot 10^{-4}$ |

Initial potential temperatures at the surface are obtained from ERA5 hourly data from 1950 to present of the ECMWF
Integrated Forecasting System (Copernicus Climate Change Service, 2019). Accordingly, in summer it is 20.7 °C, which cor-
responds to the 2 m air temperature on the 22.07.2023 at 00:00 local time for the selected coordinates. For the winter scenario
the initial potential temperature at the surface is $pt\_surface = 275.75 K$ which is 2.6 °C. The u-component (west-east) of the
wind speed is 3 m/s and the v-component (north-south) is 0 m/s. The LBC we have specified assume that there is an outflow
boundary. Therefore, a positive u-component is required to ensure that the wind does not flow out at the (left) inflow side. With
a negative u-component, the radiation boundary conditions on the right side would fail, because in this case there would be no
proper outflow in the domain. Further, we include humidity in the domain and the surface water vapor/ total water mixing ratio,
$q\_surface$ in summer, is 0.007 kg/kg, which is derived from the WRF data (for Berlin at z = 5 m). In winter it is 0.004 kg/kg.

To estimate initial soil temperatures a two-year parameterized soil spin-up is performed without the influence of an interac-
tive atmosphere, each for winter and summer. We purposely chose a long spin-up period. Thus, deeper soil layers can adapt.
Hence, at the beginning of the actual 3-D simulation the soil heat flux and temperature are very close to an equilibrium with
the atmosphere. In detail, first, all eight soil layers are horizontally homogeneous i.e., it is a horizontally isothermal soil. By
default, the soil layers have a depth (from top to bottom) of 0.005 m, 0.02 m, 0.05 m, 0.1 m, 0.2 m, 0.4 m, 0.8 m and 2.0
m. During this process the soil- and wall-layer temperatures are tailored to the prevailing atmospheric conditions. Due to the
fact that we simulate each scenario with a different land cover type, this procedure is repeated for each land cover type, which
are: bare soil, tall grass, and short grass. The land cover types are set in the according p3d file (called *vegetation_type*). In
addition, the land cover is prescribed in the static driver. The last wall/soil spin-up temperature output serves for the initial
soil temperature profile for all eight layers for the actual run, which can be seen in Table 3. In the actual simulations, the soil
spin-up runs only three days. In general, the soil spin-up is performed before the 3-D atmosphere is activated, in order to save
computational costs and time and to avoid misleading heat fluxes at the beginning of the model simulation (Maronga et al.,
2020). Table 3 also comprises the initial soil moisture (water volume per soil volume), the soil type, the root fraction, and the



*albedo_type* assigned by PALM-4U. Most of the parameters vary with the land cover and the respective depths. These data are
derived from the PALM-4U documentation page (Land surface Parameters PALM-4U).

We cannot set a fixed temperature in the soil depicting a fixed heat source, hence we accept that in the model the heat or
cold source changes its temperature with time. Thus, running the model after the soil spin-up for three days is a compromise
that gives enough time to allow the temperature signal to travel to the surface but not enough time to significantly alter the
underground heat or cold source.

**Table 3.** Initial conditions at 00:00 local time in different depths under the three land covers.

| Land Cover | zt soil | -2.0 m | -0.8 m | -0.4 m | -0.2 m | -0.1 m | -0.05 m | -0.02 m | -0.005 m |
|---|---|---|---|---|---|---|---|---|---|
| **Bare Soil** | **Summer soil temperatures** [°C] | 21.665 | 23.832 | 24.746 | 25.171 | 22.624 | 20.447 | 18.802 | 17.909 |
| | **Winter soil temperatures** [°C] | 12.501 | 6.482 | 4.614 | 3.754 | 2.569 | 1.720 | 1.106 | 0.775 |
| | **Root Fraction** | 0 | 0 | 0 | 0 | 0 | 0 | 0 | 0 |
| | **Soil Moisture** [$m^3/m^3$] | 0.3 | 0.3 | 0.3 | 0.3 | 0.3 | 0.3 | 0.3 | 0.3 |
| | **Albedo Type** | | | | 17 | | | | |
| | **Soil Type** | | | | Medium-Fine | | | | |
| **Tall Grass** | **Summer soil temperatures** [°C] | 17.224 | 18.825 | 19.380 | 19.747 | 19.469 | 19.108 | 18.787 | 18.600 |
| | **Winter soil temperatures** [°C] | 9.071 | 5.208 | 3.956 | 3.388 | 2.899 | 2.551 | 2.294 | 2.153 |
| | **Root Fraction** | 0.009 | 0.27 | 0.27 | 0.27 | 0.27 | 0.27 | 0.27 | 0.27 |
| | **Soil Moisture** [$m^3/m^3$] | 0.3 | 0.3 | 0.3 | 0.3 | 0.3 | 0.3 | 0.3 | 0.3 |
| | **Albedo Type** | | | | 10 | | | | |
| | **Soil Type** | | | | Medium-Fine | | | | |
| **Short Grass** | **Summer soil temperatures** [°C] | 17.733 | 20.014 | 20.815 | 21.369 | 20.793 | 20.112 | 19.526 | 19.189 |
| | **Winter soil temperatures** [°C] | 11.072 | 5.829 | 4.141 | 3.370 | 2.689 | 2.225 | 1.889 | 1.707 |
| | **Root Fraction** | 0.004 | 0.23 | 0.23 | 0.38 | 0.38 | 0.35 | 0.35 | 0.35 |
| | **Soil Moisture** [$m^3/m^3$] | 0.3 | 0.3 | 0.3 | 0.3 | 0.3 | 0.3 | 0.3 | 0.3 |
| | **Albedo Type** | | | | 5 | | | | |
| | **Soil Type** | | | | Medium-Fine | | | | |

The amplitude for all land cover types of the diurnal near-surface temperature variation during the spin-up phase is 5.4 K in
summer and 2.4 K for winter, which is calculated from the 2 m air temperature ERA5 data (position near Berlin, see above)
(Copernicus Climate Change Service, 2019). Regarding the processor topology we use 10 processors along x-direction of the
virtual processor grid and 5 along y-direction. The simulation is run with 50 cores and 50 tasks per node.





## 3   Results

### 3.1   Impact of the lateral boundary conditions in an idealized domain

As a first step we compare the resulting differences between cyclic and Dirichlet/radiation LBC, when the domain is subjected to identical forcing for three days (summer, bare soil, status quo scenario). Atmospheric potential air temperatures below 35 m are 10 K - 20 K warmer under cyclic LBC, with soil temperatures warming marginally due to the coupling (Fig. 2). Cyclic LBC produce larger differences between soil- and atmospheric temperatures at the interface compared to Dirichlet/radiation LBC (not shown).

In winter the behaviour of the cyclic domain is notably different. After three days of simulation time, cyclic LBC produce several degrees colder potential air temperatures during night (Fig. 2). During daytime, warmer potential air temperatures are found. While winter soil temperatures with cyclic LBC show slight cooling compared to the Dirichlet/radiation domain, this difference decreases with depth. In addition, the differences between air- and soil temperature at the interface are less pronounced in the cyclic domain as shown in Fig. 5. For Dirichlet/radiation LBC, this difference is largely unaffected by season (not shown).

When running the model for an entire year (not shown), only the Dirichlet/radiation LBC leads to a plausible annual cycle. With cyclic boundary conditions air- and soil temperatures show seasonal variations, but the domain heat up beyond 100 K. This indicates that the system accumulates energy over time and is not stable. Such accumulation is caused by the domains imitation of an infinitely large bare soil plane, with no thermal sinks other than the deep soil. Convective cooling in this case only redistributes energy, but does not dissipate it sufficiently to balance the radiative forcing. Hence, we analyze the results under cyclic LBC after three days of simulation, when this effect is not yet dominating the system.

The temperature profiles under different LBC differ not only near the surface, but in the entire modeled atmosphere up to 4000 m (Fig. 2). For Dirichlet/radiation LBC, the profile at the left (inflow) boundary throughout the entire simulation remains the same as the initial profile. This constant forcing hinders the atmosphere to develop freely. Near the surface, potential air temperatures decrease up to a height of 100 m, leading to an unstable atmosphere meaning the warmer (thinner) air is located below the colder (denser) air. The so-called inversion is located at the height of about 200 m, from thereon the potential temperature increases with height, resulting in a stable atmosphere. Due to this stable stratification, vertical motion and mixing is suppressed. The stable temperature gradient do not change substantially for larger heights. Despite the formation of diurnal cycles in the temperature profile, the prescribed profile is prominent throughout the simulation. Hence, the summer and winter potential air temperature profiles, which develop throughout the simulations with Dirichlet/radiation LBC are not useful for our analysis as they mostly reflected the values defined in the LBC.



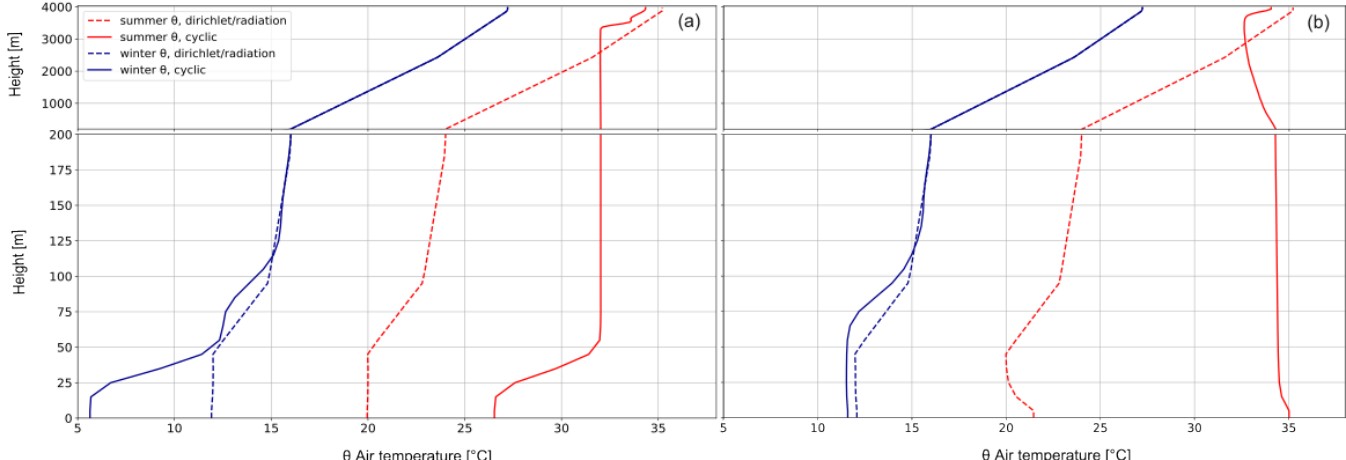

**Figure 2.** Horizontally-averaged vertical potential air temperature profiles of the summer and winter default scenario with Dirichlet/radiation and cyclic lateral boundary conditions up to a height of 4000 m over a bare soil after three days of simulation at (a) 04:00 and (b) 14:00. Note that the y-axis is not scaled uniformly.

This is in contrast to the cyclic LBC, where the profile develops more freely and produces a more neutral, even slightly unstable temperature profile, compared to the stable Dirichlet/radiation profile. In summer, the vertical temperature gradient is reduced (Fig. 2). Furthermore, a diurnal cycle develops, such that the potential air temperature decreases slightly with height, producing slightly unstable conditions up to 3.7 km. Above that, the conditions are more stable, with temperatures increasing with height. In winter, the cyclic LBC produces substantially different temperature profiles, similar to the Dirichlet/radiation profiles. These differences between summer and winter are the result of different energy flows. In summer a surplus of energy due to solar radiation leads to the soil heating up and sensible heat transferring energy to the air near the surface. Thus, the heat flux is from the ground into the atmosphere (except at night). In winter larger long-wave infrared emission from the ground produces greater cooling of the soil. In addition, less solar radiation transfers heat into the system during the day. As potential air temperatures are warmer than soil temperatures, the energy flux becomes negative, meaning that the air transfers sensible heat to the ground. The resulting stable stratification and suppression of convective mixing causes the temperature profile to behave quite similar to the Dirichlet/radiation profile.

Next, we investigate the 2-D spatial patterns of potential air temperature (exemplary for bare soil) and the wind in the domain, which maintains its westerly direction throughout the whole simulation (not shown). Under Dirichlet/radiation LBC, the u and v components correspond to the initial profile (3 m/s and 0 m/s). Under cyclic LBC, the effects of the boundary layer development are evident.





In Fig. 3 we show the instantaneous and 1 h averaged potential air temperature x-y horizontal cross sections at 5 m height for different LBC. In panel (a) instantaneous values for the Dirichlet/radiation LBC are shown. Due to the above mentioned constant forcing at the left boundary, and the homogeneous westerly flow, a gradient forms along the flow through the domain, as the air receives energy from the soil along this trajectory. The intensity of the gradient differs with height but there are no significant differences between the average and the instantaneous cross section.

In contrast, with cyclic LBC different patterns developed, although with very marginal potential air temperature differences as shown in panel (b) of Fig. 3. The hour-to-hour variations of these patterns are large. For instance, a wave pattern forms after 27 h, which shows diagonal alignment. Just an hour later fine structures are visible (not shown). In panel (c) cell structures are evident based on instantaneous data.

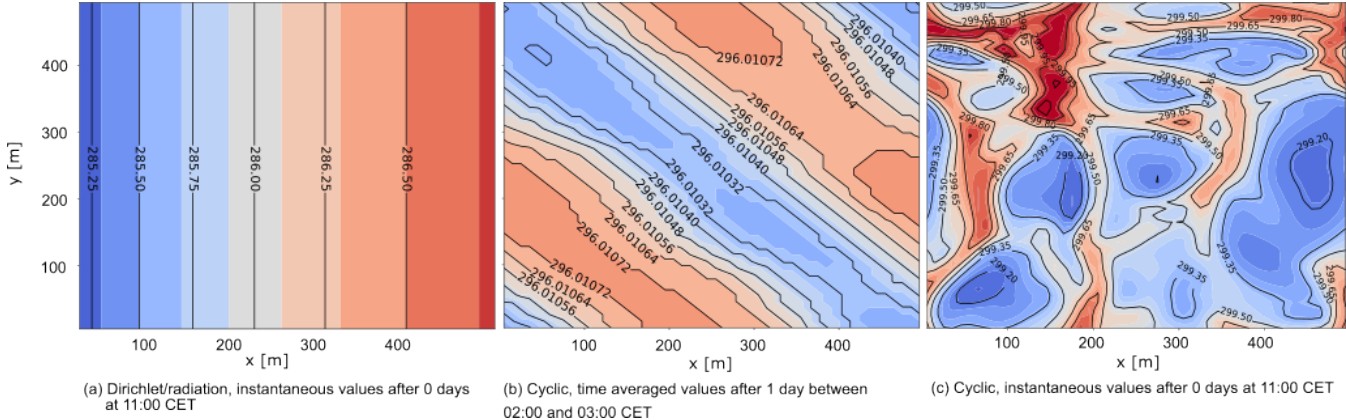

**Figure 3.** (a) Instantaneous potential air temperature at 5 m above surface over the whole domain 11 h after starting the simulation with Dirichlet/radiation boundary conditions over a bare soil; (b) time averaged potential air temperature at 5 m above surface over the whole domain between 26 and 27 h after starting the simulation with cyclic boundary conditions; (c) instantaneous potential air temperature at 5 m above surface over the whole domain 11 h after starting the simulation with cyclic boundary conditions.

An examination of the x-z cross section reveals that even under Dirichlet/radiation LBC, the lowest layers up to 35 m height
along the x-axis are not entirely laminar. However, in the higher atmosphere the stable temperature stratification is constant in x (not shown).

Under cyclic LBC, the profile is more turbulent the longer the simulations run, especially after the third day in the lower heights along the x-axis. In terms of stratification, a thick, neutral residual layer develops at nighttime and remains until 09:00 as shown in Fig. 4 (a) for 04:00 after two days. In addition, the surface cools down while in the height hotter potential air temperatures
are present. During midday (Fig. 4 (b) e.g., after 3 days at 12:00) when the soil heats up, a reversed temperature profile develops and cooler temperatures reach the higher atmosphere. In the afternoon (Fig 4 (c), e.g., after 3 days at 15:00) a thin layer of hot air at about 3800 m forms above the colder layers. Fig. 4 also reveals the expected course of the daily boundary layer height. This can also be substantiated by studying the humidity (not shown) (Hennemuth and Lammert, 2006; Kraus, 2008). In our simulation, the afternoon boundary layer height is about 1700 m. In the late afternoon at 17:00 there is hardly any gradient



and the maximum degree of mixing has been reached, such that the moisture is lifted upwards as far as possible from the ground. We find this in consecutive instantaneous humidity x-z cross section (not shown). The formed structures "disappeared" at heights around 1800 m, indicating detrainment, such that we identify this to be the boundary layer height. However, as the temperature gradient counteracts, mixing cannot take place effectively.

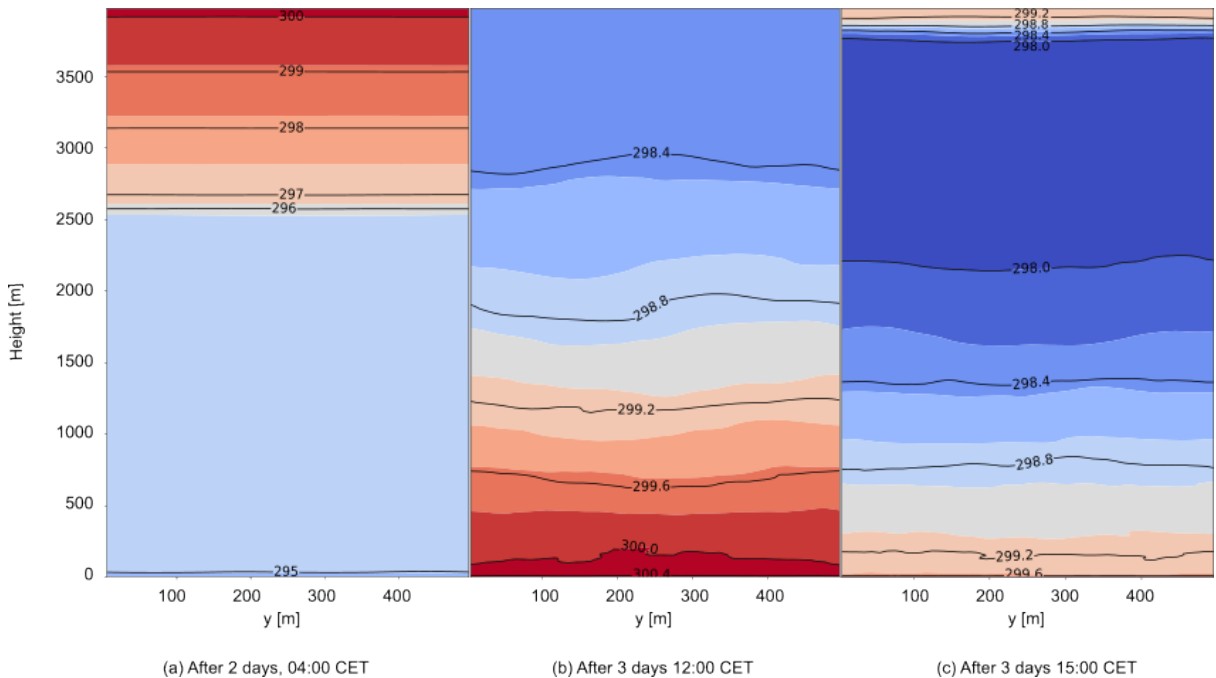

**Figure 4.** Potential air temperature time averaged (over the previous hour) x-z plots at y=250 m (centerline of the domain) up to 4000 m above ground under cyclic LBC over bare soil at different times of day during the simulation.

## 3.2   Coupling of soil and atmosphere with cyclic lateral boundary conditions

We run all simulations shown in Fig. 1 with both, Dirichlet/radiation and cyclic LBC. However, there are no effects of soil anomalies on potential air temperatures with Dirichlet/radiation LBC. Therefore, we concentrate on the changes under cyclic LBC.

### 3.2.1   Impact of the season and daytime

In Fig. 5 (a) we show the potential air temperature differences ($\Delta\Theta$) between the default bare-soil scenario and the scenario with a heat or cold source for the entire third day for summer and winter. In addition, at 04:00 and 14:00 the absolute soil temperatures and potential air temperatures up 35 m height are shown for summer and winter in panels (b)-(e). As a response to the modification of the temperature of the deepest soil layer, the thermal signal travels upwards impacting both, soil and



potential air temperatures. When deep soil temperature increases by +5 K, the potential air temperatures in the first 35 m after
3 days over a bare soil are on average 0.2 K warmer than in the default scenario in summer, and 0.3 K warmer in winter (Fig.
5 (a)). In general, the changes in potential air temperature are less pronounced in summer than in winter. The cold source
scenario produces similar but inverse patterns.

The differences between the default scenario and the modified scenarios (both increase and decrease in the deep soil temper-
ature) are greatest at 10:00 and lowest at 14:00 for the winter scenario. In summer most pronounced differences are reached
275   shortly after sunrise at 07:00.

Considering Fig. 5 (b)-(e) it is evident that the soil temperatures for the various scenarios i.e., cold source, heat source, and
the default scenario, converge towards the surface.

In addition, we find a typical day/night pattern of potential air temperatures. At night (Fig. 5 (b), (d)), potential air temperatures
rose above 15 m, while during the day (Fig. 5 (c), (e)) the air cools slightly with increasing height. Another point is that the
280   differences between the soil temperatures the closest to the surface (-0.005 m depth) and the potential air temperature closest
to the surface (5 m height) diverge the most during the day. These differences are prominent in summer due to high radiation
intensities (Fig. 5 (c)), while the soil- and potential air temperatures close to the surface do not differ strongly during the night
(Fig. 5 (b), (d)). In addition, during the day at 14:00 potential air temperatures show a rather low sensitivity to changes in the
deep soil temperature compared to the night and morning hours.





**Figure 5.** Potential air temperature modifications in 5 m height over a bare soil under cyclic boundary conditions for the whole third simulated day for winter and summer. Additionally, at 04:00 and 14:00 the potential soil- and the potential air temperature profile until a height of 35 m for the default scenario, +5 K, and -5 K deep soil temperature (-2 m) is shown.





### 3.2.2 Impact of the land cover

Studying the impact of land cover the results behave as expected: Temperatures are more sensitive as the insulation decreased (Brunsell et al., 2011). Tall grass is most inert, whereas bare soil is most sensitive. Different land covers have a significant influence on the absolute air- and soil temperatures (Fig. 6 (b)-(e), 7 (b)-(e)). Potential air temperatures over a bare soil are partly more than 10 K warmer than over tall- or short grass. In comparison, the modifications in potential air temperature based on changing deep soil temperature change only slightly for different land covers (Fig. 6 (a), 7 (a)). Additionally, in winter the potential air temperature modifications over different land covers are more similar than in summer. A change of 5 K has an effect on the potential air temperature of up to a maximum of 0.3 K for all land covers. In summer the anomalies are most pronounced for bare soil, where they are larger than 0.1 K throughout the day (Fig. 5 (a)). In contrast, short grass and tall grass show potential air temperature anomalies of less than 0.05 K during daytime (Fig. 6 (a), Fig. 7 (a)). It is also striking that, unlike over bare soil and short grass, over tall grass increasing or decreasing the deep soil temperature do not show an inverse potential air temperature profile between 17:00 and 07:00 compared to the respective other scenario. Furthermore, bare soil has a more constant modification profile over the course of the day. With tall grass, however, strong variations in the diurnal cycle are visible (Fig. 7 (a)).

Considering the fluxes, as shown in the Appendix A, a 5 K difference cause on average a 2.5 % change in the ground (soil) heat flux in the course of the third simulated day in summer for a bare soil and 5.1 % change in winter. For short grass it is 5.4 % in summer and 2.8 % in winter. Similarly, for tall grass it is 7.7 % in summer and 3.1 % in winter.





**Figure 6.** Potential air temperature modifications over short grass under cyclic boundary conditions for the whole third simulated day for winter and summer. Additionally, at 04:00 and 14:00 the potential soil- and the potential air temperature profile until a height of 35 m for the default scenario, +5 K, and -5 K deep soil temperature (-2 m) is shown.

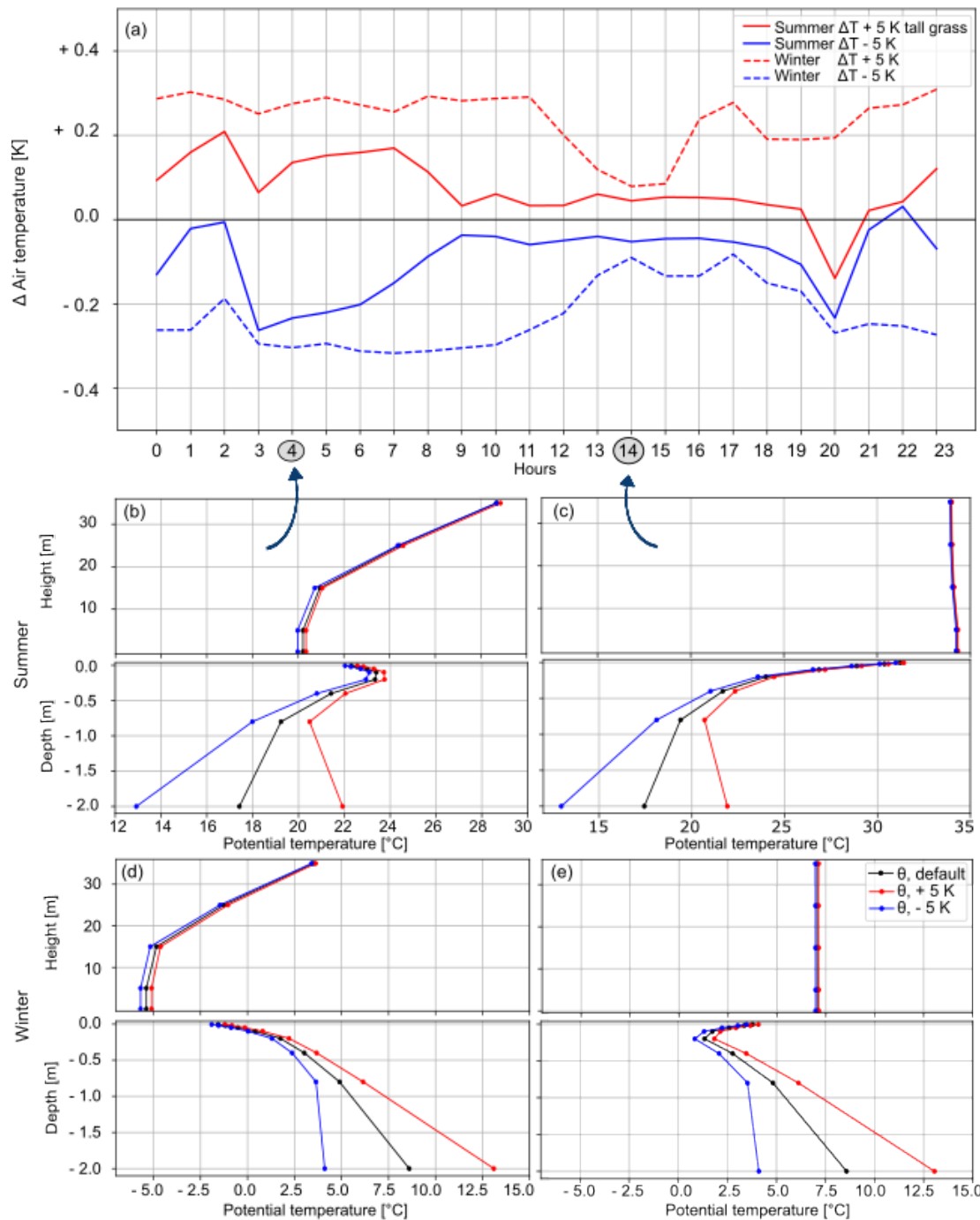

**Figure 7.** Potential air temperature modifications over tall grass under cyclic boundary conditions for the whole third simulated day for winter and summer. Additionally, at 04:00 and 14:00 the potential soil- and the potential air temperature profile until a height of 35 m for the default scenario, +5 K, and -5 K deep soil temperature (-2 m) is shown.





## 4 Discussion

### 4.1 Differences between Dirichlet/radiation and cyclic lateral boundary conditions

Under cyclic conditions the atmosphere can develop freely; in contrast with Dirichlet/radiation LBC a constant potential air
temperature profile is prescribed at the left boundary. Although the air receives energy from the surface as it moves through the
domain, this energy leaves the system through the radiation boundary on the right side and remains no longer in the domain
(Maronga et al. 2020, PALM-4U Documentation). Consequently, deep soil temperature anomalies have no detectable effect on
potential air temperatures under Dirichlet/radiation LBC.

Furthermore, the potential air temperatures under cyclic and Dirichlet/radiation conditions differ greatly due to a radiation
imbalance (Maronga et al. 2020, PALM-4U Documentation). Longer hours of sunshine in summer lead to an excess of energy
in the system, particularly under clear-sky conditions. Accordingly, under cyclic LBC energy accumulates continuously in
the system and the domain warms steadily over the course of the simulation. In addition, the outflowing air returns through
the inflow, thus small imbalances accumulate over time (Maronga et al. 2020; Schumann and Sweet 1988; Lund et al. 1998,
PALM-4U Documentation). Additionally, there is no sink of energy in the system. Thus, after just three days the temperatures
under cyclic LBC are significantly higher compared to Dirichlet/radiation LBC.

In winter temperatures are colder under cyclic LBC than with Dirichlet/radiation. The radiation deficit causes a net cooling
under cyclic conditions. The temperature becomes colder throughout the run until a stationary point is reached. Under the clear
sky radiation balance, there is no equilibrium between emission and incoming radiation. More long-wave radiation is emitted
that lead to lowering down the soil temperature. Thus, after three days the temperatures are already more than 10 K lower than
under Dirichlet/radiation LBC. In addition, the air continues to emit energy to the ground after the three days. With lower deep
soil temperatures the energy flow from the soil into the atmosphere is even lower (Wanner et al., 2022). Additionally, it must be
mentioned that the inherent reactions of soil and atmosphere lead to different time scales for adaptions to the soil temperature
anomalies. The atmosphere reacts quickly whereas the soil is an inert system (Asaeda and Ca, 1993; Benz et al., 2022; Tissen
et al., 2019; Staniec and Nowak, 2016). There is a hysteresis, the inertial effects caused by the different reaction time scales,
which causes that the system cannot be in a full balance after a short time period.

Another reasonable aspect is that with cyclic LBC the air- and soil temperature values are closer together. Generally, an
equilibrium between air and soil can establish, as the air adapts to the soil and vice versa (Gehrke et al., 2020; Maronga et al.,
2020). The longer the simulation runs, the closer the two temperatures will approach each other. Under Dirichlet/radiation
LBC, the difference between these two temperatures depends on the amount of imbalance in the forcing data.

Regarding the spatial evolution of the variables (x-y cross section, see Fig. 3) a constant gradient prevails in the domain with
Dirichlet/radiation LBC. Since the LBC are prescribed, no free development is possible in the atmosphere. The cyclic LBC
allow more freedom for the variables to develop (Schumann and Sweet, 1988). In a broader sense, the solar radiation begins
to heat the ground after sunrise and different structures form. Depending on the gradient of temperature difference, turbulent
movements arise. These are smaller in the beginning, when the solar radiation is still relatively low, but grow larger during
the day. In the x-z cross section disturbances can be seen even with Dirichlet/radiation LBC. The disturbance that enters the




atmosphere in the model originates from the deep soil. Although the stable profile strongly suppresses vertical movements, it has little effect on the near-surface potential air temperature.

## 4.2   Impact of land cover and seasonality with cyclic boundary conditions

The highest absolute temperatures and the highest offset between soil and potential air temperatures develop over a bare soil
(Fig. 5 (b)-(e)). This can be explained by a low heat capacity of bare soils and the lack of vegetation and thus evapotranspiration (Brunsell et al., 2011). Comparing all three land cover types, deep soil temperature modifications impact the potential air temperature differently, although, in a small magnitude (Fig. 5(a), Fig. 6(a), Fig. 7 (a)). However, seasonality and the time of the day have a more pronounced influence on the modifications of potential air temperature induced by our different scenarios. The effect of changing deep soil temperatures on potential air temperatures is greater in winter than in summer, and
more pronounced at night than during daytime when the atmosphere effectively transports heat upwards through convection. In winter, when temperatures are cooler, the potential air temperature increases with height i.e., the atmosphere is stable. The soil cools down and the air transfers heat to the ground and becomes cooler. Due to this stable stratification, vertical air mixing is suppressed. In contrast, in summer the air near the surface is additionally heated from the soil. With more heat in the soil there is more outgoing long wave radiation and more convection. The colder, heavier air above falls below the warmer air
due to buoyancy effects (Kraus, 2008). Furthermore, in summer, when the energy in the system increases i.e., when the deep soil temperature increases, a higher boundary layer develops (not shown). The energy is distributed over more volume. On the other hand, if the soil is cooled, the opposite effect occurs, and it stabilizes. With removing heat from the soil, convection is suppressed. With less transport of heat in the atmosphere the net transport is reduced. This effect is not linear, but exponential (Hennemuth and Lammert, 2006; Wanner et al., 2022; Kraus, 2008). Therefore, the decrease with cooling is more pronounced
than the increase with heating. The atmosphere responds more to colder temperatures. At night, the air temperature modifications are more pronounced. This can be expected as the system is dominated by the sun during the day. At night, on the other hand, there is no heat source, it is only dominated by radiative cooling.

## 5   Conclusions and recommendations for future simulations

The aim of this study is to examine the sensitivity of potential air temperatures on soil temperature alterations in unaltered, homogeneous environments, and moreover to test the simulability of idealized domains by utilizing the coupled urban microclimate model PALM-4U. The response of the idealized domain is especially dependent on the LBC. Both, Dirichlet/radiation and cyclic LBC along the x-axis have certain limitations. Only with cyclic LBC potential air temperatures are sensitive to soil temperatures. The magnitude of change depends mostly on seasonality and daytime. This amounts between 0.1 K and 0.4 K
with a change of 5 K at a depth of 2 m in the soil. Land cover has an influence on the absolute temperature as well as on the magnitude of the potential air temperature modification. However, since energy accumulates in the domain with the cyclic LBC, it should only be used for short term simulations. With the developed scenarios it was not possible to reproduce entire



realistic conditions for an ideal domain for our chosen latitude and longitude. Impacts of deep soil temperature anomalies on potential air temperatures only occur when using cyclic LBC. However, an important finding is that the time is a limiting

factor. Our recommendation is to run the model for a maximum of three days, otherwise too much energy is accumulated and temperatures become unrealistically high. With Dirichlet/radiation LBC the atmosphere cannot develop freely. Nevertheless, for using Dirichlet LBC anyway, it would be required to add a dynamic forcing from a large-scale model. One advantage is that certain physical properties and their internal consistency can be guaranteed, such as reasonable wind profiles. Thus, instead of prescribing initial constant profiles for the inlet, a dynamic driver can be provided as time-resolved, already modeled profiles.

Such a set-up even allows long-term simulations, because the energy in the system does not accumulate, as it is the case with cyclic LBC.

In summary, we show that temperature anomalies in the soil impact atmospheric potential air temperatures in some PALM-4U setups. The results underline the importance of an application-specific, elaborated and precise setting of initialization- as well as runtime parameters. In following studies we aim to transfer our findings to real-world scenarios and investigate how

the accumulation of subsurface waste heat influences atmospheric conditions. This becomes an increasingly important factor, as more interest arises in micro-climatic influences on air temperatures, especially in urban contexts.

*Code availability.* Our p3d file and static driver for running the simulation in PALM-4U 23.10 are available under: https://github.com/patikit/Glocke_PALM4U_idealized_scenarios.





## Appendix

**Figure A1.** Bare soil: Ground (soil) heat flux in summer (a), Sensible heat flux in summer (b), Ground (soil) heat flux in winter (c), Sensible heat flux in winter (d), differences of the Ground heat fluxes of the various scenarios (e), differences of the Ground heat fluxes of the various scenarios (f). Negative values represent the upwards transport from the deep soil to the surface(i.e., heat loss), positive values correspond to the flow downwards from the surface through the deeper soil layers (i.e., absorption). Be aware of the different axis labels.





**Figure A2.** Tall grass: Ground (soil) heat flux in summer (a), Sensible heat flux in summer (b), Ground (soil) heat flux in winter (c), Sensible heat flux in winter (d), differences of the Ground heat fluxes of the various scenarios (e), differences of the Ground heat fluxes of the various scenarios (f). Negative values represent the upwards transport from the deep soil to the surface(i.e., heat loss), positive values correspond to the flow downwards from the surface through the deeper soil layers (i.e., absorption). Be aware of the different axis labels.







**Figure A3.** Short grass: Ground (soil) heat flux in summer (a), Sensible heat flux in summer (b), Ground (soil) heat flux in winter (c), Sensible heat flux in winter (d), differences of the Ground heat fluxes of the various scenarios (e), differences of the Ground heat fluxes of the various scenarios (f). Negative values represent the upwards transport from the deep soil to the surface(i.e., heat loss), positive values correspond to the flow downwards from the surface through the deeper soil layers (i.e., absorption). Be aware of the different axis labels.



*Author contributions.* Conceptualization: P.G., S.A.B, B.A.K., C.C.H.; methodology: P.G., C.C.H.; validation: P.G.; formal analysis: P.G.; investigation: P.G.; writing/original draft preparation: P.G.; editing: S.A.B., C.C.H, B.A.K.; visualization: P.G.; supervision: S.A.B, C.C.H.; project administration: S.A.B. All authors have read and agreed to the published version of the manuscript.

*Competing interests.* There are no competing interests.

*Acknowledgements.* We acknowledge support by the KIT-Publication Fund of the Karlsruhe Institute of Technology. P. Glocke, S. Benz, and
B. Khan are supported by a Freigeist Fellowship, funded by Volkswagen Foundation. The authors gratefully acknowledge the computing time made available to them on the high-performance computer Horeka at the NHR Center at KIT. This center is jointly supported by the Federal Ministry of Education and Research and the state governments participating in the NHR.



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
