# Peer review of "Assessing Soil and Potential Air Temperature Coupling Using PALM-4U: Implications for Idealized Scenarios"

_EGUsphere, 2024_

## Referee Comment (RC1)

This paper investigated the impact of varying soil temperatures on potential air temperatures in an idealized domain using a model for different scenarios. This is an interesting and certainly fits into the scope of the journal. The method presented is clear, and the results are also well discussed. However, some issues presented in this investigation need further exploration. Therefore, I suggest minor revisions before accepting the paper for publication. My comments are as follows:

1. My biggest concern is why the bottom layer air potential temperature generally had so huge differences with the topsoil temperature (0.005 m depth), as shown in Figs. 5 (b-e)? The difference between the bottom layer air temperature and topsoil temperature is up to about 15 °C in Fig.5c. Because of the continuity of temperature, I think the topsoil temperature should be same or much close to the bottom air temperature, as shown in the below figure.

 Temperature

[Figure]

FIGURE 2.1. Hypothetical profiles of maximum and minimum temperature above and below soil surface on a clear, calm day.

(Reference of the figure: Campbell and Norman, 1998, An introduction to environmental biophysics, chapter 2, Page 16)

2. Please define how "Δ Air temperature" in Figs. 5-7 a was calculated.

3. Line 97, please add additional text to state how these variables are automatically prompted (or estimated).

4. In Table 2, the vertical humidity gradient is shown for the given heights. The air humidity and temperature are coupled. It would be interesting if the results of humidity profile are shown and discussed for different scenarios. How did humidity profile change for different scenarios? what the potential roles of humidity on the air temperature changes?

5. Lines 341-342, why deep soil temperature modifications impact the potential air temperature differently? Please add some potential reasons.

6. Lines 339-341, "The highest absolute temperature and the highest offset between … the lack of vegetation and thus evapotranspiration (Brunsell et al., 2011)". It seems that there is an assumption that the soil moisture of bare soil is less than that of grass soil. It is highly recommended to add additional text to clarify some assumptions.

7. Line 346, "when temperature are cooler" -> "when soil temperature are cooler".

8. Line 355, explain why "The atmosphere responds more to colder temperature".

Format:

9. Line 71, there is an edit error. Change "PALM-4U?;" ->"PALM-4U;"

10. Please maintain consistency of potential temperature units in Figures 2, 5-7. In Fig.2, the x-label is "Θ Air temperature [℃]", while the x-label is "Potential temperature [℃]" in Figs. 5-7 b-e.

11. In Figs. 5-7 b-e, the x-label is for air potential temperature and soil temperature. It is not strict by using "Potential temperature [℃]" for both soil temperature and air potential temperature.

---

## Author Comment (AC1)

**Response to Reviewer 2 for the manuscript:**

**Assessing Soil and Potential Air Temperature Coupling Using PALM-4U: Implications for Idealized Scenarios**
Manuscript number: egusphere-2024-1234

In the following text the original comments by the reviewers are given in black, our answers are blue. Line numbers refer to the unmarked manuscript (i.e., no tracked changes).

Reviewer #2 (Remarks to the Author):

In their article titled "Assessing Soil and Potential Air Temperature Coupling Using PALM-4U: Implications for Idealized Scenarios," the authors aim to answer the important question of how underground temperature extremes impact atmospheric temperatures. More specifically, the authors formulate the following three research questions:

- How to depict a realistic but idealized domain in PALM-4U?
- Do heat or cold extremes in the soil modify potential air temperatures?
- What parameters affect these modifications?

The authors are moving in a new direction by investigating the effect of subsurface temperature extremes on air temperatures (and not vice-versa). In the introduction, the authors succinctly address the relevance of this novel perspective. With its interdisciplinary view of the interactions between multiple spheres of the Earth system, this research is of interest to the scientific community and beyond, making it a suitable contribution to ESD. The authors present a well-written and concise manuscript in most parts and provide a good introduction to the general relevance of the topic as well as typical approaches and limitations in understanding and implementing a thermal coupling of the subsurface and the atmosphere.

**Reply**: We thank the reviewer for their careful, comprehensive and constructive feedback which helped us to improve the manuscript significantly. Thank you for (also) highlighting the strength of the paper, this is very much appreciated.

However, I find the manuscript difficult to follow for the following reasons: First, the introduction provides a good overview of the relevance and state of knowledge, but it seems decoupled from the rest of the manuscript. For example, I assume the choice of boundary conditions is self-evident to the authors, but it may not be to every interested reader. In that way, a brief explanation of boundary conditions used in temperature simulation at interfaces (e.g., in atmospheric research) would help the reader understand the choice of boundary conditions, their pros and cons, and why they were considered for the investigation.

**Reply:** We have addressed your concerns in the following way:

The decoupling between the introduction and the rest of the manuscript: We believe the reason for this was that the introduction mainly discussed the second part of our results, the coupling of subsurface and atmosphere and only briefly mentioned the challenges we address in the first part of the results section: how to model this coupling in an idealized domain. This has now been addressed by adding a more detailed problem description and by explaining (the importance of) boundary conditions and their meaning in idealized and realistic domains. We added a new paragraph on page 2, line 70:

[In our study we ask the reverse: do alterations in soil temperatures impact potential air temperatures?]. "Due to a lack of usable real-world data, this study approaches this question numerically in an idealized domain. As such it is intended as a proof of concept, laying the groundwork for future research. Idealized domains are not yet defined in PALM-4U. Before conducting experiments, it is essential to thoroughly understand and characterize the processes in our "area of investigation".

as well on page 6, line 133:

"Within the PALM-4U model there are several options for LBCs such as Dirichlet, cyclic, those mentioned in conjunction with one radiation boundary, Neumann, turbulence re-cycling, etc., which can be looked up here: (https://palm.muk.uni-hanno-ver.de/trac/wiki/doc/app/initialization_parameters#bc) (Initialization Parameters). Further, the detailed explanation of the LBCs and how they calculate the flow is given here: (https://palm.muk.uni-hannover.de/trac/wiki/doc/tec/bc) (Boundary Conditions). We decided using the Dirichlet/radiation LBC and entire cyclic LBCs because these options are plausible for our use case. In this way the system can unfold without allowing too many degrees of freedom. The advantages and disadvantages we faced with those options are depicted in the discussion."

Second, there is a very detailed presentation/explanation and interpretation (rather than discussion) of the obtained results. My impression is that the authors focus on these highly detailed results, but the general evaluation of the model performance/suitability is not prominently discussed and/or limited to the plausibility check of the results. Therefore, the question arises of how the results and model performance can be evaluated and checked. Can available datasets or data from experiments be used to validate the results?

Reply: As of right now our numerical model cannot be validated to data from experiments as there is no data on the impact of subsurface heat or cold anomalies on atmospheric temperatures. Because of the complexity of atmospheric temperatures and air movements and the many diverse drivers of local climates, such data is more or less impossible to achieve in an experimental setup. Hence, as an initial test of our hypothesis (i.e., can subsurface heat or cold sources impact atmospheric temperatures?) we decided to focus on a quasi-idealized experiment. This is in no way meant to represent the real world, but rather intended to answer questions in the fundamental sciences, using well established numerical models to do so. We are currently working on a numerical model of a real-life domain, which will give more insight into how subsurface urban heat islands may contribute to atmospheric urban heat. However, we believe that this first step is significant enough to both climate sciences and geosciences warrant its own publication. This point was also added in the manuscript (discussion 4.1):

"A general evaluation of the model performance to check quality of the digital representation of reality cannot be assessed as there are no reliable observational data facing the question how heat sources in the soil affect ground-level atmospheric temperature. Thus, our purpose was to test our hypothesis for the first time using quasi-idealized experiments."

The very technical nature and a high degree of detailed explanation of the results make it difficult to follow the common theme and line of argumentation and really understand the work's contribution and novelty – also regarding the accuracy of the results and validity beyond the model domain. The guidance of the reader is missing or at least not apparent to me. Hence, I encourage the authors to revise the manuscript accordingly (moderate to major revision required).

Minor comments include:

1. Lines 20-27: Check the syntax for suitability in a scientific journal

**Reply**: As non-native speakers it is unclear to us what you are specifically referring to. Would you be able to point out examples? Thank you very much in advance.

2. Line 41: How relevant is the impact of an individual construction? The accumulation is probably relevant, and the examples given appear somewhat random.

**Reply**: We changed the manuscript to clarify that not only accumulation but individual constructions alone can impact groundwater and soil temperatures significantly.

"For example, ground temperatures near underground parking garages can be up to 10 K warmer. This appears within and outside the urban environment and is an addition to the accumulation of urban waste heat".

4. Line 46: Delete "However"

**Reply**: Done: "The thermal coupling between the underground and the atmosphere is complex."

5. Line 64: Please specify "near surface atmosphere"

**Reply**: Done in the manuscript: "… (until 4000 m height but with a special focus on the lowest 35 m) …"

6. Line 87: What about heat transport via percolating water? Should be addressed.

**Reply**: Advective heat transport is typically not considered dominant for subsurface heat transport, particularly in an urban environment where sealed surfaces are omnipresent. We added a short paragraph about this in the manuscript.

"Particularly in an urban environment the advective heat transport can be neglected due to the wide occurrence of sealed surfaced."

7. Line 158 and others: Is "right side" the best terminology to refer to the orientation in the model?

**Reply**: Thank you for this comment, indeed it is confusing. We changed left/right to inflow and outflow boundary in the text.

8. Line 197: You mention Fig. 2 and then Fig.5. Should Fig. 5 then be renamed as Fig. 3?

**Reply**: Very good point. Due to our storyline, we could not move Figure 5 forward. We have now changed Figure 2 in a way that we added the soil temperatures already at this point. Thus, the offset is already visible here.

---

## Author Comment (AC2)

**Response to Reviewer 1 for the manuscript:**

**Assessing Soil and Potential Air Temperature Coupling Using PALM-4U: Implications for Idealized Scenarios**
Manuscript number: egusphere-2024-1234

In the following text the original comments by the reviewers are given in black, our answers are blue. Line numbers refer to the unmarked manuscript (i.e., no tracked changes).

Reviewer #1 (Remarks to the Author):

This paper investigated the impact of varying soil temperatures on potential air temperatures in an idealized domain using a model for different scenarios. This is an interesting and certainly fits into the scope of the journal. The method presented is clear, and the results are also well discussed. However, some issues presented in this investigation need further exploration. Therefore, I suggest minor revisions before accepting the paper for publication. My comments are as follows:

**Reply**: Thank you very much for your constructive feedback. We implemented most of your suggestions which improved our research a lot. Please find detailed answers to all of your questions below.

1. My biggest concern is why the bottom layer air potential temperature generally had so huge differences with the topsoil temperature (0.005 m depth), as shown in Figs. 5 (b-e)? The difference between the bottom layer air temperature and topsoil temperature is up to about 15 ℃ in Fig.5c. Because of the continuity of temperature, I think the topsoil temperature should be same or much close to the bottom air temperature, as shown in the below figure.

[Figure]

FIGURE 2.1. Hypothetical profiles of maximum and minimum temperature above and below soil surface on a clear, calm day.

(Reference of the figure: Campbell and Norman, 1998, An introduction to environmental biophysics, chapter 2, Page 16)

**Reply**: Unfortunately, we do not have access to the book you mention here. Based on the graphic it seems to describe a very idealized but unrealistic scenario. We know from many (experimental and theoretical) studies that the offset between near surface air temperatures is highly dependent on landcover but has an offset most of the time. One of the most well-known studies is Cermak et al. 2017; They conducted measurements of air, near-surface, and shallow ground temperatures under bare soil, sand, short-cut grass and asphalt. The shallow soil temperature was generally warmer than the near surface air temperature particularly under high solar radiation (as is the case in our model). We added the following text to the manuscript on page 19 after line 340:

"The modeled high difference between soil and air temperature is in line with existing literature. For example, Cermak et al. 2017 conducted measurements of air, near-surface, and shallow ground temperatures under bare soil, sand, short-cut grass and asphalt, and found that soil temperature was generally warmer than the near surface air temperature, particularly for high solar radiation. They also found different behavior for different land cover types showing the highest offsets in summer days under asphalt."

References:

Cermak, V., Bodri, L., Kresl, M., Dedecek, P., and Safanda, J.: Eleven years of ground–air temperature tracking over different land cover types, 37, https://doi.org/10.1002/joc.4764, publisher: John Wiley & Sons, Ltd, 2017.

2. Please define how "Δ Air temperature" in Figs. 5-7 a was calculated.

**Reply**:    Thank you, we added this information in the caption of figure 5 (page 14). "… Δ Air temperature is the difference between the default scenario and the modified one.  ..."

3. Line 97, please add additional text to state how these variables are automatically prompted (or estimated).

**Reply**: We added this to the manuscript (after line 97):

"Variables like roughness lengths, emissivity, and leaf area index are automatically prompted by the model and are summarized following look-up tables: https://palm.muk.uni-hannover.de/trac/wiki/doc/app/iofiles/pids/static/tables        (Static        tables)        and https://palm.muk.uni-hannover.de/trac/wiki/doc/app/land_surface_parameters (Land Surface Parameters)."

4. In Table 2, the vertical humidity gradient is shown for the given heights. The air humidity and temperature are coupled. It would be interesting if the results of humidity profile are shown and discussed for different scenarios. How did humidity profile change for different scenarios? what the potential roles of humidity on the air temperature changes?

**Reply**: It is indeed a valuable point you mentioned. We also considered to investigate this question but have decided against focusing on it in this study. We are aware that there is a link between temperature and humidity. It is not within the scope of this study to look at the interaction with humidity. But, since we have no saturated zone, we assume that our results will not change much. The careful investigation of latent heat energy flux components is one of our future objectives.

5. Lines 341-342, why deep soil temperature modifications impact the potential air temperature differently? Please add some potential reasons.

**Reply**: We added the following explanation to the manuscript (line 341 ff.).

"Furthermore, comparing all three land cover types, deep soil temperature modifications impact the potential air temperature differently, although, in a small magnitude ((Fig. 5 (a), Fig. 6 (a), Fig.7 (a)) due to the e.g., different surface properties like heat conductivity, heat capacity, soil moisture, different surfaces energy balances and the dependent influence of the ground heat flux etc."

6. Lines 339-341, "The highest absolute temperature and the highest offset between … the lack of vegetation and thus evapotranspiration (Brunsell et al., 2011)". It seems that there is an assumption that the soil moisture of bare soil is less than that of grass soil. It is highly recommended to add additional text to clarify some assumptions.

**Reply**: Thank you, we clarified that in the manuscript (line 339 ff.).

"This can be explained by lack of vegetation, resulting in less evapotranspiration and an decreased latent heat flux. This in turn leads to an increased ground heat flux as well as decreased soil moisture and a low heat capacity of bare soils."

7. Line 346, "when temperature are cooler" -> "when soil temperature are cooler".

**Reply**: 7. We changed it to overall temperatures, because we refer to both, air and soil.

Line 346: "In winter, when overall temperatures are cooler, the potential air temperature increases with height i.e., the atmosphere is stable."

8. Line 355, explain why "The atmosphere responds more to colder temperature".

**Reply**: We added the following information in the manuscript (after line 355):

"In addition, the response to colder temperatures is greater because the energy difference is distributed over a smaller volume in the cold scenario due to the mixing depth in the boundary layer. Due to increases in mixing layer heights with seasonal changes in stability and solar irradiance, the same energy differences are distributed over a comparably larger volume during summer, resulting in local air temperature differences being less pronounced."

Format:

9. Line 71, there is an edit error. Change "PALM-4U?;" ->"PALM-4U;"

**Reply**: We apologize for the misunderstanding and reformatted our text to make things more clear

"Accordingly, we set out to conduct a sensitivity analysis and address three distinct questions:

   • (a) how to depict a realistic but idealized domain in PALM-4U?

   • (b) … "

10. Please maintain consistency of potential temperature units in Figures 2, 5-7. In Fig.2, the x-label is "Θ Air temperature [℃]", while the x-label is "Potential temperature [℃]" in Figs. 5-7 b-e.

**Reply**: Thank you for drawing this to our attention. We adapted all figures accordingly and now use the term Potential air temperature.

11. In Figs. 5-7 b-e, the x-label is for air potential temperature and soil temperature. It is not strict by using "Potential temperature [℃]" for both soil temperature and air potential temperature.

**Reply**: 11. Very good point. We changed it to Soil- and potential air temperatures. As an example, see the following figure:

[Figure]

---

## Author Response (AR2)

Thank you very much for your feedback. We appreciate it a lot and implemented your comments.

---

## Author Response (AR3)

Thank you very much for your feedback. We appreciate it a lot and implemented your comments.

We have reviewed the language in detail and improved the suggested passages as suggested by the reviewer, which was:

"I would ask the authors to double-check the language of their amendments, examples are "soil- and potential air temperature" (Fig. 7 x-axis label) should read "soil and potential air temperature", "to check quality of the…" (line 321) should read "to check the quality of the…", "like heat conductivity…" (line 368) should read "like thermal conductivity…"."

We have uniformly removed the hyphen before soil and air temperature, as this was not used consistently throughout the manuscript.

In detail we changed the following:

**Line 17:** These results help to enhance our understanding of the coupling between soil **\remove{-}** and atmospheric temperatures …

**Line 182**: During this process the soil **\remove{-}** and wall-layer temperatures …

**Line 211:** In addition, the differences between air **\remove{-}** and soil temperature at the interface are less pronounced …

**Line 215:** With cyclic boundary conditions air**\remove{-}** and soil temperatures show seasonal variations, …

**Line 296:** These differences are prominent in summer due to high radiation intensities (Fig. 5 (c)), while the soil **\remove{-}** and potential air temperatures …

**Line 303:** Different land covers have a significant influence on the absolute air**\remove{-}** and soil temperatures …

**Line 319:** A general evaluation of the model performance to check **\add{the}** quality of the digital representation of reality cannot be assessed …

**Line 344:** Another reasonable aspect is that with cyclic LBC the air**\remove{-}** and soil temperature values are …

**Line 366:** Furthermore, comparing all three land cover types, deep soil temperature modifications impact the potential air temperature differently, although, in a small magnitude (Fig. 5 (a), Fig. 6 (a), Fig. 7 (a)) due to the different surface properties like **\remove{heat} \add{thermal}** conductivity, heat capacity, soil moisture, different surfaces energy balances and the dependent influence of the ground heat flux etc.

**Caption of figure 5, figure 6, figure 7:** Additionally, at 04:00 and 14:00 the potential soil **\remove{-}** and the potential air temperature profile